# Mobbing and Violence at Work as Hidden Stressors and Work Ability among Emergency Medical Doctors in Serbia

**DOI:** 10.3390/medicina56010031

**Published:** 2020-01-13

**Authors:** Dragan Nikolić, Aleksandar Višnjić

**Affiliations:** 1Center for Healthcare Analysis, Planning and Organization, Institute of Public Health of Niš, 18000 Niš, Serbia; 2Department of Social Medicine and Public Health, Faculty of Medicine, University of Niš, 18000 Niš, Serbia

**Keywords:** mobbing, violence, work ability, emergency, medical doctors

## Abstract

*Background and Objectives:* People employed in emergency medical services represent a professional group which encounters events beyond ordinary human experience, great work demands, the risk of professional disputes, and stressful situations. The goal of this study is to examine the presence of mobbing and violence at work, as well as their influence on work ability of emergency medical doctors. *Materials and Methods:* The survey is conducted in Emergency Medical Service (EMS) in Niš in the period between December 2017 and January 2018. Using standardized questionnaires on psychosocial conditions in work environment (COPSOQ II) and work ability index (WAI) this study encompasses 79 doctors. For estimation of the examined factors’ influence on WAI linear regression analysis was used. *Results*: EMS doctors were exposed to abuse in 30.4% of the cases. The decline in WAI is significantly related with exposure to violence by patients (β = 0.727), exposure to physical violence (β = 0.896), exposure to abuse several times (β = 0.691) and exposure to ill-treatment by patients (β = 0.750). *Conclusion*: The results indicate that in the examined doctors mobbing and workplace violence are very much present and have a negative impact on their work, and therefore on the quality of health care.

## 1. Introduction

Regarding rapid alterations in the work character, risk factors at the workplace constantly change and the impact of damaging psychosocial and organizational factors on employees increases. “The new stressors” are the characteristics of modern work conditions appearing in various forms such as: abuse at the workplace, authority and duty misuse, intimidation, political party membership preference, and gender inequality. At the same time, previously determined stressors change their connotation. As the modern society of changes is governed by the unsparing struggle for better work and professional positions so as to acquire material interest, in merciless competition and global economic crisis environment, suitable conditions for the bloom of “Social Diseases” in the form of physical, emotional abuse, harassment, terror, trauma, i.e., victimization in relation to work or mobbing as well as the violence at the workplace appear. The enlisted phenomena are well known in organizational pathology in the last 30 years [1,2,3,4] as dysfunctional forms of behavior and communication.

Systematic activities on creating a healthy and secured work environment for the purpose of securing the respect for human dignity and personal integrity of employees, represent for quite a while, the priority of European and other international organizations. Sadly, so far not one official international definition of abuse at the workplace has been accepted and very few members of the EU have adopted the laws which directly prevent these issues. More and more frequently it is acknowledged that psychological abuse is often repetitive in a form, which is by its nature, maybe, insufficiently distinctive, yet which by its constant repetition turns into a serious form of violence [2,5,6].

Even though an isolated incident may cause consequences, psychological abuse is often comprised of repetitive, unwanted, violent activities performed by one party whose goal or repercussion may inflict damage to human rights and dignity, violate physical and mental health, or discredit the professional future of a victim [5]. Violence at work is manifested by incidents in which employees are unjustly exploited, subjected to physical threats and insults, or they are exposed to insulting activities of any kind at their workplaces [7].

Serbia is one of a few countries in Europe which enacted a special Abuse Prevention Act at Work [8] which penalizes the repercussions of aggressive behavior of individuals in companies and institutions. Although the problem of abuse and violence is known as much as the desire of individuals or groups for authority, dominance, and superiority at work and in everyday life, the most fertile ground for the appearance of mobbing are institutions and work environments which recognize a strict hierarchical structure with prominent culture of careerism [6].

The latest data indicate that those employed in health care and education (14.6%), administration and defense (11.6%), transport, communication, commerce, and hotel business (around 9%) are the most exposed to physical threats [9]. It is important to say that these factors negatively influence general life and work ability, [10] the quality of life of employees [11], as well as competitiveness of a work organization [1,12].

Therefore, prevention and termination of the aforementioned threats is the major task of entrepreneurs and company managers, as well as all other employees and society in general [3,4,13]. The aim of this study is to examine the impact of mobbing and violence at work, as well as the greatest causes of professional stress and its influence on working ability of emergency medical doctors in Niš.

## 2. Materials and Methods

The survey regarding mobbing and violence at work as well as their impact on the work ability was conducted including 79 doctors of medicine of the Institute for Emergency Medical Service (EMS) in Niš in the period between December 2017 and January 2018.

Participation in the survey was voluntary and anonymous with the prior announcement about the significance of the research and the approval of the management of the work organization. The study procedures were carried out in accordance with the Declaration of Helsinki. The Ethical Committee of the Faculty of Medicine, University of Niš (14-5785-3) approved the study. Written informed consent was obtained from all participants after the goals of the study and handling of collected data were explained to ensure privacy and confidentiality was understood.

For all the respondents included in the research, a special poll was made which contained: overall data, a questionnaire on psychosocial conditions in the work environment, health and welfare (COPSOQ) [14,15,16], and a questionnaire for the evaluation of the work ability (WAI) [17]. The average time for providing the poll answers was 30 min.

Overall data referred to age, gender, overall and specific years of work, marital status, lifestyle, additional jobs, shift work, and night shift working.

Questionnaire on psychosocial working conditions, health and welfare (Copenhagen Psychosocial Questionnaire—COPSOQ) is a special measuring instrument developed at Psychosociology Department of the National Institute of Occupational Health in Copenhagen, Denmark. For the purpose of proving validity of COPSOQ [14] questionnaire, numerous researches including various occupations [15] confirmed its significance for evaluation and advancement of psychosocial factors at work. An advanced version of this instrument named COPSOQ II (Short version) was used in this research containing 40 questions with 23 dimensions, intended for the use at workplaces [18]. The scale measuring COPSOQ was formed by adding the points for each question. In most of the cases there are five offered answers rated from 0 to 4. The value of the scale is calculated as arithmetic average. The new scale for evaluation of the workplace, among other things, includes the questions regarding offensive behavior at work (mobbing, violence, attacks) [18].

Work Ability Index Questionnaire (WAI) is a standardized instrument of the Finnish Federal Institute for Occupational Safety and Health [17]. It is used for examination of the work ability in relation to work demands. The work ability index (WAI) is calculated pursuant to the instructions of the authors and expressed numerically in the range from 7 to 49 points. Higher score shows better work ability. According to the number of points, WAI is ranked in four categories: bad (7–27), good (28–36), very good (37–43), and outstanding (44–49). WAI is a great prognostic indicator of the remaining employees at one specific workplace. In workers with bad WAI, there is a great risk of leaving their jobs in the next five years.

For examination of the differences between the groups of respondents on continuous variables, we used *t*-test for individual samples. Frequency comparison of some categories of distinguishing marks was done by Fischer test of the exact probability of the Null Hypothesis in certain cases where one of the expected mark frequencies was lower than five.

For evaluation of the influence of the examined factors on WAI, we used linear regression analysis. Coefficients of linear regression (β) are calculated and displayed along with their 95% confidence intervals. Evaluation of the statistical significance of the value was performed by *t*-test. Coefficients represent changes in WAI caused by the value increase independent of variables for one unit of measurement. The analysis started with the application of univariate (simple) regression models. Afterwards, the multivariate models were formed by the back-step method, where all of the factors which failed to show a significant influence on WAI were being excluded. The statistical analysis was performed using the SPSS 17.0 program (SPSS Inc., Chicago, IL, USA) in Windows 7 Ultimate. The statistical significance was set at *p* < 0.05.

## 3. Results

Socio-demographic and occupational characteristics of the respondents are shown in the Table 1. Out of 79 doctors examined from Emergency Medical Service (EMS), 50 (63.3%) of them were women and 29 (36.7%) were men. The highest percentage belonged to the age groups 40–49 and 50–59 years of age (32.9% each), and the lowest percentage was represented by respondents aged 60 and over (2.5%). The majority of doctors (83.5%) from EMS have changeable working hours during night shifts (Fischer test: *p* < 0.001).

The mean value of the work ability index (WAI) of medical doctors employed in EMS was 38.30 ± 1.40. Working ability in relation to physical demands is bad in 1 (1.3%), good in 14 (17.7%), very good in 28 (35.4%), and outstanding in 36 (45.6%) respondents, while in relation to mental health requirements are bad in 1 (1.3%), good in 6 (7.6%), very good in 17 (21.5%), and outstanding in 55 (69.6%) respondents.

Simple regression analysis confirmed that age was significantly associated with WAI values. Physicians younger than 30 had a WAI value of 1.081 higher (95% CI = 0.185–1.977; *p* = 0.018) than subjects from all other age groups, while physicians aged 50–59 had a WAI value of 0.789 less (95% CI = 0.296–1.281; *p* = 0.002) than respondents from all other age groups.

Simple linear regression analysis confirmed that the following dimensions of the COPSOQ questionnaire were significantly associated with WAI values: Emotional demands, Commitment to the workplace, Predictability, Rewards (recognition), Role clarity, Job satisfaction, Work-family conflict, Self rated health, Burnout and Stress. Each increase in the value of the next dimension scores by 1 was associated with a significant increase in the WAI value (Table 2).

Of the job characteristics and scores of COPSOQ questionnaires, multiple regression analysis identified Role clarity and Self rated health as the most significant factors influencing WAI values (Table 3). Each increase in scores by 1 was associated with an increase in WAI for values of Role clarity by 0.166 (95% CI = 0.022–0.311; *p* = 0.025) and Self rated health by 0.600 (95% CI = 0.409–0.790; *p* < 0.001). The regression model containing these factors explains 35.3% of the variability in the WAI (coefficient of determination R^2^ = 0.353).

The analysis of data acquired from questionnaire portions referring to evaluation of psychosocial conditions at work (COPSOQ II short), which are related to mobbing and violence at work indicates that doctors from EMS are exposed to unwanted sexual attention in 12.7% (Table 4). In addition, 38 doctors (48.1%) in urgent medicine were exposed a number of times to violence threats over the past year. However, doctors from EMS are exposed to threats by the subordinates, but more frequently displayed by patients or their relatives (45.6%).

The majority of doctors from EMS were, more than once, victims of physical violence over the year, and it was mostly displayed by patients (Table 4). The results indicate that 30.4% of medical doctors from EMS experienced mobbing throughout the year, where in a significant percentage, from the abuse manifested by patients or their relatives.

Based on simple linear regression analysis, it has been confirmed that the following parameters are noticeably related to increased values of WAI: absence of violence threats (β = 0.714), absence of physical violence (β = 0.811) and abuse at the workplace (β = 0.652), and shifts mainly between 06:00 and 18:00 (β = 0.613) (Table 5).

In contrast, variable working hours without night shifts (β = 2.376), along with violence threats by patients (β = 0.727), multiple annual physical violence (β = 0.896), and annually repeated mobbing (β = 0.691), as well as abuse by patients or their relatives (β = 0.750) are in relation to a considerable decrease of WAI (Table 5).

Afterwards the multivariate model was formed by the back-step method, where all the factors which failed to show a significant influence on WAI were being excluded. By means of multiple linear regression analysis, out of the characteristics showing the presence of violence and abuse at work, we single out inexposure to violence threats (β = 0.646) as the factors with a positive influence on WAI, and multiple annual physical aggression incidents (β = –0.746) as the negative ones. A regression model which contains the aforementioned factors explains 9.0% of the variability of WAI value (determination coefficient R^2^ = 0.090) (Table 3).

## 4. Discussion

The aim of medical education, being at the same time education for a lifetime, is to breed doctors in possession of a wide range of knowledge and skills for the purpose of maximum patient care. A doctor’s job implies great psychosocial demands, constant contacts with patients, colleagues, and other people. At the same time, the people who have chosen this calling are expected to display 24 h professionalism, regardless of their mood, character, or biorhythm [19]. On the other hand, numerous papers more often report about frequent health disturbances [20], mental symptoms [21], and burnouts [22] in the people employed in the health care department.

Among professional risk factors in the work environment of doctors, for the emergency service doctors in particular, it is important to pick out negative organic reactions to prolonged stress, emotional exhaustion, and the influence of intolerance syndrome to shift work which is more evident in the cases of fatigue and exhaustion [19]. A significant problem that is often overlooked or not recognized and rarely discussed is abuse and violence by patients, health service users, colleagues and associates.

Serbia, as a transitional country, goes through numerous economic and political changes; therefore the expected hierarchy of damaging psychosocial factors in the work environment and stressors at work is completely different. The healthcare sector was forced to change and improve the quality of services provided and at the same time to lower the costs throughout last fifteen years. Due to a false image of social welfare, expressed in the form of an inadequate and insufficiently reasonable concept “health care users entitled to full rights” created by powerful mass media in an environment where it is of paramount importance to work on health promotion and a culture of health improvement, poverty changes the definition of tolerance and trust, especially primary care physicians and thus negate the goals previously set and doom the users as well as all the participants in providing health care to “hidden loss”.

Previous researches, which did not encompass health care workers, pointed out that the quality of occupational life could have a positive impact on productivity level [23], the risk of occupational injuries [24], and that professional stress, often used as a theoretical framework for defining the quality of life in relation to work, is related to diversity of negative physical, psychological, and behavioral effects [25]. In that sense, evaluation of psychosocial conditions represents an indicator as important as a good practice guidance or evidence-based medicine. Researches in this field provide priceless data and help in recognizing working conditions of the people employed in health department, particularly those in the ’’front line’’. Various indicators, such as early retirement, frequent cardiovascular, musculoskeletal and malignant diseases, anxiety-depression disorders and post-traumatic stress disorder, increased mortality, and sudden death [26] suggest that EMS employees are at a greater risk than the overall working population and the people employed in other health sector. All this is related to an influence of special conditions and work demands not studied enough in this profession [26], and a potentially dangerous effect of the exposure to everyday labor critical incidents [21,27,28]. The data from recent literature [2,5,6] show the necessity to deal with this problem due to lack of knowledge on stress prevalence, types of stressful situations and their impact on health while performing this specific service, as well as all the positive activities and screening interventions, effective preventive measures, and activities which have to be organized and conducted. Apart from that, this is one of the possible ways to resolve some current problematic issues which are of paramount significance for health care system development in Serbia [3,4,13].

Even though mobbing and violence are separate dimensions of professional existence, in our surroundings they defy scientific and professional conceptualization. Very few research studies have dealt with this problem in Serbia, mostly talking about nursing jobs [29]. Efforts of the authors of this paper about the appearance of mobbing and its repercussions on health and welfare of EMS employees during 2006–2007 did not provide relevant results due to a very small number of answers from the part of the poll referring to mobbing (11.6%) [29]. Compared to our present findings, there is only one logical conclusion indicating that the enactment of the Abuse Prevention Act at Work in Serbia had a great influence on surpassing fear, distrust, and suspense of employees about the reason for being surveyed, by whom and in what purpose the obtained data might be used [8].

It should be stated that violence and disturbance at work are noticeable psychological risks in the health sector in EU countries. Health and social services had the majority of reports on violence at work in EU-27 (15.2%), and the incidence was above average. Violent behavior was displayed 8 times more in the health department (where physical assaults were previously rare), in relation to occupations where physical assaults are more frequent (manufacture and construction work), where the attackers were mostly patients, colleagues, and visitors [9]. In a Danish project called Violence as a Form of Expression, 32% of social workers and nurses in hospitals were the victims at work [30]. In the latest British research on aggression and violence at work in the National Health Center, it was established that between 29–50% of medical workers and ambulance drivers were exposed to violence over the last 12 months [7]. Researches conducted over the past few years in neighboring countries show that 76% of doctors had mobbing experience in the last year, which is more than the number reported in previous studies [31]. Even though psychosocial stimuli have been depicted as “subjective creation of an individual mind” for quite some time, and the reports about their influence were recently made, it is unquestionable that abuse and violence at work have a considerable impact on professional actions of health care workers [19,31].

According to the studies of Polish authors, 62% of doctors who work in a first aid stations were intimidated by the patients, while 11% of them experienced physical assault [19]. In the Czech Republic 12% of doctors that work in the health and social sector experienced physical assaults, especially those who work with patients or clients. Doctors who work in EMS are more exposed to threats and violence then those who work in other health institutions. In 17% of the cases, verbal aggression (shouting and insulting) is experienced by superiors, 6% by colleagues, and 3% by patients [6].

The connection between shift work and chronic fatigue is not clearly determined due to the fact that the latter cannot be attributed only to the shift-working conditions [32]. On the contrary, the connection between shift work and sleep deprivation is better documented [33]. Even though there are some individual differences often ascribed to increased vigilance rates, shift work can deepen fatigue and increase probability of mistakes at work [34,35]. Recent studies [36,37] indicate that sleep, health, and safety disorders are in a way correlated with professional stress, improper nutrition, and decreased physical activity. Lack of sleep can lead to a number of diseases such as cardiovascular conditions, diabetes, and mental disturbances including burnouts [33,34]. Generally, when talking about productivity evaluation, circadian rhythm and sleep disorders [37] can be significant factors along with a dose-dependent response in relation to sleep length [38], although individual differences caused by the impact on productivity decrease remain an important issue [37].

Working ability is closely connected with good health and depends on professional knowledge as well as organizational, ergonomic, and psychosocial conditions at work. Inability to control and lack of free time at work [39,40] can affect WAI. Doctors who work at EMS have decreased WAI because of conflicts between work and family responsibilities and stress at work. All these psychosocial factors mentioned above are more dominant with doctors that work in urgent medical service.

### Limitations and Strengths

The most significant limitation of the study was the sample size. This is because the survey was conducted in only one institution. Nevertheless, all physicians employed there have participated at that time. However, it is also an advantage. Namely, in this paper we “scanned” the whole institution, which is one of the most representative of its kind in Serbia.

In addition, researchers used a self-report method for this data collection. Therefore, it would be possible that a certain number of respondents had the desire to “reduce their issues” and therefore overestimate their experiences (possible bias).

This research adds to the scientific literature invaluable experiences from Serbia, where health services are (still) available to all citizens. The city of Niš is the second largest city in the Republic of Serbia (historically it was also a capital), and its EMC best reflects the whole issue of EMC in the whole country. Otherwise, it is one of the few centers in Serbia where EMC operates as an independent institution (not within any of the Health Centers). Our research has shown that in the future, emphasis should be placed on preventing mobbing and abuse at workplace more than ever, because, among other things, the quality of health care is also declining.

## 5. Conclusions

In Serbian’s health care facilities, mobbing and violence at work must be further systematically researched. Health care workers are at greater risk than other occupations. Although violence and maltreatment at work cannot be easily measured, there are many things that can change the working environment. Mobbing and violence at work increases costs and has a negative effect on productivity in work organizations. Despite the small number of data concerning mobbing at work, it is certain that it affects working ability and can lead to mental health disorders. For this reason preventive interventions are needed to promote mental and physical health and welfare. Strategies to reduce psychosocial work stressors should be included in quality management programs to maintain EMC doctors’ work ability.

## Figures and Tables

**Table 1 medicina-56-00031-t001:** General characteristics of the respondents.

Characteristic	EMS (*n* = 79)
**Sex**	
Female	50 (63.3%)
Male	29 (36.7%)
**Age**	
<30 years	7 (8.9%)
30–39 years	18 (22.8%)
40–49 years	26 (32.9%)
50–59 years	26 (32.9%)
>60 years	2 (2.5%)
**Marital status**	
Married	60 (75.9%)
Live together	4 (5.1%)
Live with parents	8 (10.1%)
Divorced	3 (3.8%)
Singles	4 (5.1%)
**With how many children live**	
None	17 (21.5%)
1	32 (40.5%)
2	28 (35.4%)
3 or more	2 (2.5%)
**Weekly work hours (h)**	40.29 ± 3.58
**Paid overtime work (h)**	0.77 ± 2.86
**Unpaid overtime work (h)**	2.38 ± 6.73
**Other jobs (h)**	0.41 ± 1.93
**Work in shifts**	
Primarily between 06:00 and 18:00	9 (11.4%)
Primarily between 22:00 and 06:00	1 (1.3%)
Variable hours without night work	3 (3.8%)
Variable opening hours with night work	66 (83.5%)
**Internship at the institution (years)**	13.68 ± 9.65

**Table 2 medicina-56-00031-t002:** Assessing the association of the Copenhagen Psychosocial Questionnaire (COPSOQ) scores by its dimensions with work ability index (WAI) values, results of simple linear regression analysis.

Scores	Dimensions	β	95% CI		t	*p*
HMP (*n* = 79)			Lower	Upper		
1.131.26	Quantitative demands	−0.071	−0.248	0.105	−0.798	0.426
4.92 ± 1.78	Work pace	0.095	−0.027	0.218	1.539	0.126
5.18 ± 1.67	Emotional demands	−0.224	−0.328	−0.121	−4.279	<0.001
5.97 ± 1.89	Influence	0.106	−0.008	0.220	1.832	0.069
5.97 ± 1.38	Possibilities for development	0.101	−0.047	0.249	1.355	0.178
7.25 ± 1.19	Meaning of work	−0.015	−0.172	0.142	−0.185	0.853
4.82 ± 1.34	Commitment to the workplace	0.310	0.160	0.460	4.082	<0.001
5.24 ± 1.49	Predictability	0.147	0.006	0.288	2.055	0.042
4.66 ± 1.57	Rewards (recognition)	0.163	0.026	0.301	2.354	0.020
6.75 ± 1.31	Role clarity	0.248	0.079	0.416	2.910	0.004
4.99 ± 2.15	Quality of leadership	0.101	−0.006	0.208	1.859	0.065
5.71 ± 2.10	Social support from supervisor	0.051	−0.059	0.161	0.916	0.361
2.10 ± 0.49	Job satisfaction	0.456	0.031	0.882	2.122	0.036
3.59 ± 1.78	Work−family conflict	−0.202	−0.322	−0.083	−3.341	0.001
5.33 ± 1.42	Trust regarding management	0.151	−0.009	0.312	1.866	0.064
4.32 ± 1.74	Justice and respect	0.019	−0.101	0.140	0.319	0.751
2.00 ± 1.00	Self rated health	0.704	0.513	0.894	7.310	<0.001
4.70 ± 1.82	Burnout	−0.283	−0.390	−0.175	−5.190	<0.001
4.85 ± 1.54	Stress	−0.221	−0.341	−0.101	−3.651	<0.001

Notes: β—Beta coefficient in regression ANOVA analysis of potential predictors; CI—Confidence interval.

**Table 3 medicina-56-00031-t003:** Results of the multiple linear regression analysis assessing the relationships between COPSOQ scores and WAI.

Characteristics	β	95% CI		*t*	*p*
		Lower	Upper		
**COPSOQ scores (dimensions 1–19)**					
Role clarity	0.166	0.022	0.311	2.273	0.025
Self rated health	0.600	0.409	0.790	6.219	<0.001
**COPSOQ scores (dimensions 20–23)**					
Inexposure to threats of violence	0.646	0.179	1.113	2.736	0.007
Exposure to physical violence	−0.746	−1.494	−0.003	−1.991	0.049

Notes: β—Beta coefficient in regression ANOVA analysis of potential predictors; CI—Confidence interval.

**Table 4 medicina-56-00031-t004:** Mobbing and violence in the workplace COPSOQ Questionnaire scores (by domain 20–23) for sexual harassment, threats of and physical violence, and abuse at the workplace.

Characteristics	EMS (*n* = 79)
**Sexual harassment in the last 12 months**	
Every day	-
Once in a week	-
Once in a month	1 (1.3%)
Only few times	9 (11.4%)
No	69 (87.3%)
**Unwanted attention was expressed by**	
Colleague	3 (3.8%)
Manager	-
Subordinate	4 (5.1%)
Patients/clients	4 (5.1%)
**Exposure to threats of violence in the last 12 months**	
Every day	3 (3.8%)
Once in a week	4 (5.1%)
Once in a month	4 (5.1%)
Only few times	27 (34.2%)
No	41 (51.9%)
**Threats are directed by**	
Colleague	2 (2.5%)
Manager	1 (1.3%)
Subordinate	-
Patients/clients	36 (45.6%)
**Exposure to physical violence in the last 12 months**	
Every day	-
Once in a week	1 (1.3%)
Once in a month	-
Only few times	12 (15.2%)
Not at all	66 (83.5%)
**Violence is manifested by**	
Colleague	1 (1.3%)
Manager	-
Subordinate	-
Patients/clients	13 (16.5%)
**Exposure to abuse in the last 12 months**	
Every day	1 (1.3%)
Once in a week	2 (2.5%)
Once in a month	1 (1.3%)
Only few times	20 (25.3%)
Not at all	55 (69.6%)
**The abuse was manifested by**	
Colleague	4 (5.1%)
Manager	4 (5.1%)
Subordinate	3 (3.8%)
Patients/clients	15 (19.0%)

**Table 5 medicina-56-00031-t005:** Assessing the relationship between work characteristics, sexual harassment, threats of and physical violence, and abuse at the workplace and WAI: a simple linear regression analysis.

Characteristic	β	95% CI		t	*p*
		Lower	Upper		
**Weekly working hours (h)**	−0.015	−0.055	0.025	−0.725	0.469
**Paid overtime work (h)**	0.028	−0.064	0.120	0.600	0.549
**Unpaid overtime (h)**	−0.019	−0.060	0.022	−0.907	0.366
**Other jobs (h)**	0.003	−0.047	0.052	0.112	0.911
**Work experience in HMP (years)**	−0.011	−0.036	0.013	−0.899	0.370
**Working primarily between 06:00 and 18:00**	0.613	0.172	1.054	2.748	0.007
**Working primarily between 22:00 and 06:00**	0.346	−2.316	3.007	0.257	0.798
**Variable opening hours without night work**	−2.376	−3.870	−0.881	−3.143	0.002
**Variable opening hours with night work**	−0.420	−0.868	0.028	−1.853	0.066
**Sexual harassment**					
Exposure to sexual harassment—monthly	−0.662	−3.322	1.998	−0.492	0.624
Exposure to unwanted sexual attention—few times a year	0.209	−0.563	0.982	0.535	0.593
Inexposure to sexual harassment	−0.143	−0.891	0.604	−0.379	0.705
Exposure to unwanted sex. att.—colleague	0.353	−0.991	1.698	0.520	0.604
Exposure to unwanted sex. att.—subordinates	0.359	−0.747	1.465	0.642	0.522
Exposure to unwanted sex. att.—patients	−0.474	−1.680	0.732	−0.778	0.438
**Exposure to threats of violence**					
Exposure to threats of violence—daily	−0.677	−2.018	0.665	−0.998	0.320
Exposure to threats of violence—weekly	−0.934	−2.271	0.402	−1.382	0.169
Exposure to threats of violence—monthly	−0.677	−2.018	0.665	−0.998	0.320
Exposure to threats of violence—few times a year	−0.506	−1.029	0.017	−1.914	0.058
Inexposure to threats of violence	0.714	0.247	1.181	3.022	0.003
Exposure to threats of violence—colleagues	−0.331	−1.878	1.217	−0.423	0.673
Exposure to threats of violence—manager	−0.662	−3.322	1.998	−0.492	0.624
Exposure to threats of violence—subordinates	−0.162	−1.508	1.184	−0.238	0.813
Exposure to threats of violence—patients/clients	−0.727	−1.218	−0.236	−2.929	0.004
**Exposure to physical violence**					
Exposure to physical violence—weekly	0.346	−2.316	3.007	0.257	0.798
Exposure to physical violence—few times a year—patients	−0.896	−1.654	−0.138	−2.337	0.021
Inexposure to physical violence	0.811	0.076	1.547	2.182	0.031
Exposure to physical violence—colleague	−0.667	−2.553	1.219	−0.699	0.486
**Exposure to abuse**					
Exposure to abuse—daily	−0.159	−2.049	1.730	−0.167	0.868
Exposure to abuse—weekly	−0.162	−1.508	1.184	−0.238	0.813
Exposure to abuse—monthly	−0.662	−3.322	1.998	−0.492	0.624
Exposure to abuse—few times a year	−0.691	−1.241	−0.141	−2.487	0.014
Inexposure to abuse	0.652	0.144	1.160	2.540	0.012
Exposure to abuse—colleagues	−0.542	−1.567	0.483	−1.045	0.298
Exposure to abuse—manager	−0.169	−1.040	0.701	−0.384	0.701
Exposure to abuse—subordinate	−0.241	−1.269	0.788	−0.463	0.644
Exposure to abuse—patients/clients	−0.750	−1.425	−0.075	−2.196	0.030

Notes: β—Beta coefficient in regression ANOVA analysis of potential predictors; CI—Confidence interval.

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
