# Peer review of "Mobbing and Violence at Work as Hidden Stressors and Work Ability among Emergency Medical Doctors in Serbia"

_1010-660X, 2020, doi:10.3390/medicina56010031_

Round 1

Reviewer 1 Report

Thanks for the opportunity to review the manuscript.

The theme of the present article is of interest, but the novelty of this research is questionable.

A number of previous papers investigated similar relationship in larger populations, finding a clear influence of mobbing and violence at the workplace on the work ability

(e.g. Fischer FM & Martinez FC Work. 2013;45(4):509-17. doi: 10.3233/WOR-131637.

Fischer Fm et al. Chronobiol Int. 2006;23(6):1165-79 DOI: 10.1080/07420520601065083.

Martinez MC et al. Am J Ind Med. 2015 Jul;58(7):795-806. doi: 10.1002/ajim.22476)

I can find no element of novelty in your research, nor a literature gap you want to fill with your study design.

Moreover, the simple size is too small to generalize your results.

Average value of WAI is surprisingly very low (8,3) in your sample of medical doctors. How can you explain this result?

I suggest to report results on WAI calssification (percentage of bad - medium - high category) 

The use of statistical terms such as ANOVA, MANOVA and linear regression is confusing. An in-depth review of the statistical methods is needed. I expected to found crude and adjusted beta coefficients and the variance explained by the model.

Conclusions are not supported by the results. 

You stated that violence at work affect "quality of work, absence from work, decrease of the quality of social living and functioning, early retirement and the loss of experts at their most productive age", but you didn' measure any of these varaibles. How can you draw this conclusion?

Limitations of the study must be reported.

State clearly what this paper adds to the scientific literature in the field.

Author Response

Dear sirs,

Almost all your requests, suggestions and objections have been accepted and amendments have been made accordingly.
Please look again at the whole paper.
Significant changes are colored red.
Thanks for your attention.

Reviewer 2 Report

Dear Authors,

I really appreciate your work. I have some suggestions and comments to improve it. See below.

Page 2 and troughout all the manuscript: pay attention to term used. For example, replace 'abuse at the work post' with 'abuse in workplace' or similar. Please, insert a definition of mobbing.

Method section: what about ethics guideline? and privacy statement, how did you guarantee the anonimity? I suggest you to insert the ethic statement (see Declaration of Helsinki World Medical Association. (2001). World Medical Association Declaration of Helsinki. Ethical principles for medical research involving human subjects. Bulletin of the World Health Organization79(4), 373).

About the scales used, I suggest to insert examples of items (for each scale, in particular for mobbing, violence and so on).

Discussion section: What about limitation of your research? Please, include limitation and explain possible bias.

Conclusion section: I suggest to insert here some practical strategy to prevent/intervene to improve the well-being of your sample.

Author Response

(The authors gave the same response as above.)

Round 2

Reviewer 2 Report

Dear Authors,

I appreciate your efforts to improve the manuscript according to comments and suggestions offered.